# Therapeutic Effects of Quetiapine and 5-HT_1A_ Receptor Agonism on Hyperactivity in Dopamine-Deficient Mice

**DOI:** 10.3390/ijms23137436

**Published:** 2022-07-04

**Authors:** Yukiko Ochiai, Masayo Fujita, Yoko Hagino, Kazuto Kobayashi, Ryoichi Okiyama, Kazushi Takahashi, Kazutaka Ikeda

**Affiliations:** 1Addictive Substance Project, Tokyo Metropolitan Institute of Medical Science, Tokyo 156-8506, Japan; ochiai-yk@igakuken.or.jp (Y.O.); fujita-ms@igakuken.or.jp (M.F.); hagino-yk@igakuken.or.jp (Y.H.); 2Department of Neurology, Tokyo Metropolitan Neurological Hospital, Tokyo 183-0042, Japan; ryouichi_okiyama@tmhp.jp (R.O.); kazushi_takahashi@tmhp.jp (K.T.); 3Department of Molecular Genetics, Institute of Biomedical Sciences, Fukushima Medical University, Fukushima 960-1295, Japan; kazuto@fmu.ac.jp

**Keywords:** psychiatric symptoms, dopamine deficiency, dopamine-deficient mice, 5-HT_1A_ receptor, Parkinson’s disease

## Abstract

Some diseases that are associated with dopamine deficiency are accompanied by psychiatric symptoms, including Parkinson’s disease. However, the mechanism by which this occurs has not been clarified. Previous studies found that dopamine-deficient (DD) mice exhibited hyperactivity in a novel environment. This hyperactivity is improved by clozapine and donepezil, which are used to treat psychiatric symptoms associated with dopamine deficiency (PSDD). We considered that DD mice could be used to study PSDD. In the present study, we sought to identify the pharmacological mechanism of PSDD. We conducted locomotor activity tests by administering quetiapine and drugs that have specific actions on serotonin (5-hydroxytryptamine [5-HT]) receptors and muscarinic receptors. Changes in neuronal activity that were induced by drug administration in DD mice were evaluated by examining Fos immunoreactivity. Quetiapine suppressed hyperactivity in DD mice while the 5-HT_1A_ receptor antagonist WAY100635 inhibited this effect. The number of Fos-positive neurons in the median raphe nucleus increased in DD mice that exhibited hyperactivity and was decreased by treatment with quetiapine and 5-HT_1A_ receptor agonists. In conclusion, hyperactivity in DD mice was ameliorated by quetiapine, likely through 5-HT_1A_ receptor activation. These findings suggest that 5-HT_1A_ receptors may play a role in PSDD, and 5-HT_1A_ receptor-targeting drugs may help improve PSDD.

## 1. Introduction

Dopamine is a neurotransmitter that plays a very important role in motor control, motivation, reward, and cognitive function [1]. When dopamine levels in the brain decrease, basal ganglia circuits, including the striatum, become imbalanced, resulting in parkinsonism, characterized by bradykinesia, rigidity, resting tremor, and postural instability [2,3]. Such drugs as metoclopramide and cerebrovascular disorders, tumors, and inflammatory and infectious processes that involve areas of the nigrostriatal pathway can decrease dopamine in the brain and cause parkinsonism [4]. Dopamine levels also decrease in Parkinson’s disease (PD) through the degeneration of dopaminergic neurons in the substantia nigra, the cause of which has not yet been clarified [5].

Some diseases that are associated with dopamine deficiency are accompanied by psychiatric symptoms [6]. Manganese has been reported to accumulate in dopamine neurons in the ventral tegmental area and substantia nigra [7] and is known to cause secondary parkinsonism and psychiatric symptoms, such as hallucinations [8]. Hallucinations have also been reported in patients with pathologically proven vascular parkinsonism [9]. Cases of brain tumors with parkinsonism and hallucinations have also been reported [10]. Furthermore, an inverse correlation was found between a reduction in striatal dopamine transporters and neuropsychiatric symptoms of PD [11]. However, the precise pathogenesis of psychosis with dopamine deficiency has not been clarified.

Drugs that have been reported to effectively treat PSDD include clozapine, pimavanserin, quetiapine, and cholinesterase inhibitors [12,13]. Although the efficacy of quetiapine has reportedly varied [13], it is still used clinically because it is safe, does not exacerbate motor symptoms, and does not have strong side effects. However, the mechanisms of action of these drugs remain to be elucidated.

DD mice are genetically modified mice that are unable to synthesize dopamine. Dopamine depletion in these mice occurs through the knockout of the tyrosine hydroxylase gene. The subsequent loss of epinephrine and norepinephrine can be prevented by restoring tyrosine hydroxylase expression under control of the dopamine β-hydroxylase promoter [14,15]. Dopamine-deficient mice are maintained with regular L-DOPA supplementation. When the L-DOPA dosage is reduced, these mice exhibit bradykinesia that mimics clinical symptoms of parkinsonism. However, when dopamine levels in the brain are excessively depleted and fall below the limit of detection, DD mice become hyperactive, which is detected by an increase in locomotor activity in a novel environment [16]. Why DD mice become hyperactive in a novel environment is unknown, but a decrease in acetylcholine may be involved. These DD mice continuously run around a new environment, jump sometimes, and exhibit excited behaviors without becoming acclimated to their environment. This hyperactivity can be ameliorated by clozapine and donepezil (cholinesterase inhibitor). Hyperactivity in mice can be recognized as a psychiatric symptom [17,18]. Likewise, hyperactivity in DD mice may reflect abnormal psychiatric behaviors.

PSDD and hyperactivity in DD mice occur in dopamine-deficient states and is improved by clozapine and donepezil. Thus, we considered that DD mice could be used to study PSDD. In the present study, we investigated the pathogenesis of PSDD and mechanism of action of drugs that improve PSDD in DD mice. We first administered drugs in DD mice that exhibited hyperactivity, reflecting PSDD, and analyzed whether these drugs suppress hyperactivity. Next, we investigated the mechanism of action of drugs that improve hyperactivity in DD mice. We found that quetiapine effectively reduced hyperactivity in DD mice via serotonin 5-hydroxytryptamine-1A (5-HT_1A_) receptor stimulation. Quetiapine and the 5-HT_1A_ receptor agonist 8-hydroxy-2-dipropylaminotetralin hydrobromide (8-OH-DPAT) suppressed the hyperactivity-associated increase in Fos expression in the median raphe nucleus (MRN). These results suggest that 5-HT_1A_ receptor stimulation may suppress activity in the MRN and inhibit hyperactivity in DD mice either directly or indirectly by suppressing 5-HT release or affecting other neurotransmitter systems.

## 2. Results

### 2.1. Quetiapine Ameliorated Hyperactivity in DD Mice

We first examined the effects of drugs on hyperlocomotion in DD mice. To induce hyperactivity, DD mice with brain dopamine levels that were below the limit of detection were placed in a novel environment, and locomotor activity was recorded for 3 h. The mice were then injected with saline, 20 mg/kg quetiapine, pimavanserin, tandospirone, paroxetine, or trihexyphenidyl, and locomotor activity was continuously recorded for another 3 h. We calculated locomotor activity by subtracting locomotor activity counts that were measured in the first 3 h (before drug administration) from locomotor activity counts that were measured in the second 3 h (after drug administration; Figure 1). Negative numbers and positive numbers indicated a decrease and increase, respectively, in locomotor activity in the second 3 h. Locomotor activity decreased because of environmental habituation in the second 3 h in saline-treated wildtype (WT) mice. Saline-treated DD mice exhibited hyperlocomotion. These results are consistent with our previous study [16]. We found that quetiapine suppressed locomotor activity in both WT and DD mice. The other drugs, including pimavanserin, tandospirone, paroxetine, and trihexyphenidyl, did not significantly affect locomotor activity in either WT or DD mice. The time course of locomotor activity in saline- and 20 mg/kg quetiapine-treated mice indicated that quetiapine significantly suppressed locomotor activity in both WT and DD mice beginning 20 min after administration until the end of the test. Data for quetiapine doses of 10, 40, and 80 mg/kg are included in the Appendix A. The transient increase in activity in WT mice just prior to drug administration may be attributable to the fact that the mice may have detected that the researchers had entered the laboratory and administered the drugs to the other mice. The increase in activity immediately after drug administration may be attributable to the pain of the injection. Conversely, in DD mice, excitation was transiently suppressed immediately after drug administration (Figure 2a,b).

### 2.2. 5-HT_1A_ Receptor Antagonist Partially Inhibited the Effect of Quetiapine

In a previous study, clozapine was shown to be effective against hyperactivity in DD mice, and the mechanism was considered to involve an increase in acetylcholine [16]. Quetiapine is a Multi-Acting Receptor-Targeted Antipsychotic (MARTA) like clozapine, but it is known to have minimal actions on acetylcholine receptors. To further investigate the mechanism of action, we administered the 5-HT_1A_ receptor antagonist WAY100635 (1 or 0.2 mg/kg) and anticholinergic drug scopolamine (0.1 mg/kg) 30 min before 20 mg/kg quetiapine administration to examine whether these receptors mediate the locomotor-suppressive effect of quetiapine. The effects of quetiapine were partially blocked by 1 mg/kg WAY100635 (Figure 3a). When 0.2 mg/kg WAY100635 was administered, the effect was not as clear (Appendix A). Scopolamine did not significantly block the effects of quetiapine (Figure 3b). In WT mice, the effects of quetiapine were not blocked by WAY100635 or scopolamine (Appendix A). The effects of quetiapine are presumed to be partially mediated by the 5-HT receptor system.

### 2.3. 5-HT_1A_ Receptor Agonist Ameliorated Hyperactivity in DD Mice

The effects of quetiapine were partially mediated by 5-HT_1A_ receptors, suggesting that 5-HT_1A_ receptor function might be involved in hyperactivity in DD mice. Quetiapine also targets 5-HT_2A_ receptors. Thus, receptor subtype-specific drugs were administered to determine which specific receptor subtypes may be involved in hyperactivity in DD mice. 8-OH-DPAT (5-HT_1A_ receptor agonist) and EMD281014 (5-HT_2A_ receptor antagonist) were administered in WT and DD mice. 8-OH-DPAT was more effective than EMD281014 in suppressing hyperactivity (Figure 4a,b).

### 2.4. Number of Fos-Positive Cells was Reduced by Quetiapine and 8-OH-DPAT in the MRN

We performed immunohistochemical staining using an anti-c-fos antibody to examine changes in activated brain regions that are related to the 5-HT system when hyperactivity was suppressed by quetiapine or 8-OH-DPAT. Fos-positive cells were counted in a 400 μm × 100 μm area in the MRN and 200 μm × 200 μm area in the rostral linear nucleus (RLi). Hyperactive DD mice exhibited an increase in Fos-positive cells in the MRN, which was reduced by quetiapine and reduced further by 8-OH-DPAT (Figure 5). In contrast, few Fos-positive cells were found in the RLi (Figure 6). No Fos-positive cells were detected in the dorsal raphe nucleus (data not shown).

## 3. Discussion

In the present study, we found that quetiapine, a drug that is used to treat psychotic symptoms of PD [13], effectively suppressed hyperactivity in DD mice.

Although quetiapine is a MARTA, it does not target muscarinic receptors. As expected, we confirmed that scopolamine did not block the effects of quetiapine. Therefore, hyperactivity should be inhibited through other molecular targets beyond muscarinic receptors. Among multiple receptors that are targeted by quetiapine, the present study focused on 5-HT receptors. WAY100635 partially inhibited the effects of quetiapine, suggesting that 5-HT_1A_ receptors participate in mediating the effects of quetiapine. Quetiapine stimulates 5-HT_1A_ receptors and inhibits 5-HT_2A_ receptors. The 8-OH-DPAT and EMD281024 results indicate that the inhibition of hyperactivity was mainly mediated by 5-HT_1A_ receptor stimulation. Therefore, the results of the present study infer a 5-HT_1A_ receptor-mediated mechanism that underlies the effects of quetiapine on PSDD. In addition to providing clues to the mechanism of action of quetiapine on PSDD, 5-HT_1A_ receptors may be a target for effectively treating PSDD.

5-HT_1A_ receptors are a major mediator of the actions of 5-HT. The 5-HT_1A_ receptor is a metabotropic Gprotein-coupled receptor that is highly expressed in 5-HT neurons as a presynaptic inhibitory autoreceptor. 5-HT_1A_ receptors are expressed in many brain regions that are innervated by 5-HT neurons, including the frontal cortex, septum, amygdala, hippocampus, and hypothalamus, as postsynaptic heteroreceptors [19]. We tested two 5-HT_1A_ receptor agonists in the present study. 8-OH-DPAT but not tandospirone mitigated hyperactivity in DD mice. Tandospirone acts on postsynaptic 5-HT_1A_ receptors [20,21]. In contrast, 8-OH-DPAT is a full 5-HT_1A_ receptor agonist [22,23]. The present results suggest that the presynaptic function of 5-HT_1A_ receptors may be important for the inhibition of hyperactivity in DD mice. Previous studies demonstrated the importance of the presynaptic function of 5-HT_1A_ receptors rather than their postsynaptic functions for the treatment of a low level of social interaction, anxiety, and cognitive dysfunction [23,24]. Thus, presynaptic 5-HT_1A_ receptors may also play an important role in improving PSDD.

Baseline 5-HT levels are elevated in the striatum in DD mice [16]. The early disruption of central dopaminergic pathways is known to increase striatal 5-HT content [25]. The stimulation of presynaptic 5-HT_1A_ receptors results in a decrease in 5-HT release. Therefore, high 5-HT levels may be a cause of hyperactivity, and the suppression of these high levels may be a mechanism by which hyperactivity is reduced. In humans, 5-HT neurons are degenerated in PD [26]. The degeneration of 5-HT neurons occurs more slowly than the degeneration of dopaminergic neurons, which may result in a 5-HT-dominant state that is a common pathological feature of PSDD and hyperactivity in DD mice.

Psychiatric symptoms have been studied in other DD animals. DD animals that exhibit psychiatric symptoms include 1-methyl-4-phenyl-1,2,3,6-tetrahydropyridine (MPTP)-lesioned marmosets [27], human α-synuclein transgenic rats [18], and 6-hydroxydopamine-lesioned rats [28]. However, psychotic behaviors appeared when these animals received drugs that stimulate dopamine receptors. Therefore, these models may not be suitable for studying the pathogenesis of psychosis in a dopamine-depleted state. With regard to 5-HT and psychosis, L-DOPA-stimulated psychosis-like behavior in MPTP-lesioned monkeys was abolished after 5-HT injury by 3,4-methylenedioxy-*N*-methamphetamine (MDMA) [29]. Although the animals in this previous study received L-DOPA, the fact that serotonergic neuron injury can improve neuropsychiatric-like behaviors in DD animals appears to support the present results, in which a decrease in serotonergic signaling could ameliorate psychotic-like behavior in DD animals.

We found that the number of Fos-expressing neurons significantly increased in the MRN in DD mice that exhibited hyperactivity. In contrast, few Fos-positive cells were detected in the RLi. The MRN is a major nucleus of 5-HT neurons. However, some 5-HT neurons and other neurons release other neurotransmitters, such as γ-aminobutyric acid (GABA) and glutamate [30]. Interestingly, the MRN mediates motor activity through both the agonism and antagonism of several neurotransmitter receptors, including GABA, glutamate, and opioid receptors [30]. Therefore, any of these neurons in the MRN may be involved in hyperactivity in DD mice. Future studies will determine which neurons are specifically activated in hyperactive DD mice.

One limitation of this study is that c-Fos is an indirect marker of neuronal activity because it is expressed after action potentials spike in neurons [31]. The appearance of c-Fos-positive cells that is attributable various nonspecific causes should also be considered. Future studies, such as calcium imaging, will be required to analyze neuronal activity more precisely. Additionally, both DD mice and wildtype mice exhibited a decrease in locomotor activity in the novel environment when they were treated with quetiapine and 8-OH-DPAT. Therefore, it is unknown whether quetiapine and 8-OH-DPAT specifically suppressed hyperlocomotion in DD mice.

## 4. Materials and Methods

### 4.1. Drugs

We used L-DOPA (Sigma Aldrich, St. Louis, MO, USA), ascorbic acid (Sigma Aldrich), benserazide hydrochloride (FujiFilm, Osaka, Japan), MediGel (Clear H_2_O, Westbrook, ME, USA), DietGel (Clear H_2_O), sodium barbiturate (Nacalai Tesque, Kyoto, Japan), pimavanserin (5-HT_2A/2C_ receptor inverse agonist; Toronto Research Chemicals, Toronto, ON, Canada), paroxetine (selective 5-HT reuptake inhibitor; Tocris Bioscience, Bristol, UK), tandospirone (5-HT_1A_ receptor agonist; Tocris Bioscience), quetiapine (MARTA; Toronto Research Chemicals), EMD281014 (5-HT_2A_ receptor antagonist; Tocris Bioscience), 8-OH-DPAT (5-HT_1A_ receptor agonist; Sigma Aldrich), WAY100635 maleate (5-HT_1A_ receptor antagonist; Sigma Aldrich), clozapine (Toronto Research Chemicals), scopolamine (muscarinic receptor antagonist; Nacalai Tesque), oxotremorine-M (muscarinic receptor agonist; Sigma Aldrich), and trihexyphenidyl (muscarinic receptor antagonist; Tokyo Chemical Industry, Tokyo, Japan).

L-DOPA for injection was dissolved in ascorbic acid solution to 1.4 mg/mL and administered at 50 mg/kg. Ascorbic acid solution was made by dissolving ascorbic acid in saline at a concentration of 2.5 mg/mL. Pimavanserin (5 mg/kg), tandospirone (3 mg/kg), quetiapine (10, 20, 40, and 80 mg/kg), and EMD281014 (10 mg/kg) were dissolved in one-tenth the required amount of dimethylsulfoxide and diluted to the final volume with saline or purified water. Clozapine (10 mg/kg) was dissolved in a minimum volume of 0.1 N HCl and diluted to the required volume in saline or purified water. Paroxetine (8 mg/kg), 8-OH-DPAT (10 mg/kg), WAY100635 (0.2 and 1 mg/kg), scopolamine (0.1 mg/kg), oxotremorine-M (0.1 mg/kg), and trihexyphenidyl (3 mg/kg) were dissolved in saline or purified water.

### 4.2. Dopamine-Deficient Mice

Dopamine-deficient mice were created as described previously [14]. We used DD mice (*n* = 10–29) and wildtype (WT) mice (*n* = 10–21), which were littermates that were created by crossing heterozygous/heterozygous DD mice on a C57BL/6J genetic background. The experimental procedures and housing conditions were approved by the Institutional Animal Care and Use Committee (Animal Experimentation Ethics Committee, Tokyo Metropolitan Institute of Medical Science, approval no. 12–43). All of the animals were cared for and humanely treated according to our institutional animal experimentation guidelines. All of the mice were housed in an animal facility that was maintained at 23 °C ± 1 °C and 55% ± 5% relative humidity under a 12 h/12 h light/dark cycle (lights on at 8:00 AM, lights off at 8:00 PM). Food and water were available ad libitum. Because newborn rats cannot eat sufficient food that contains L-DOPA during the neonatal period, L-DOPA was administered intraperitoneally 6 days per week until the DD mice reached 6 weeks of age. Afterward, the mice were given paste food or MediGel that was supplemented with L-DOPA in addition to their usual food pellets. We examined male and female mice at 10–58 weeks of age.

The paste food that was supplemented with L-DOPA was prepared by mixing 1000 mg of L-DOPA, 500 mg of ascorbic acid, and 250 mg of benserazide in 2 kg of powdered food. The paste was created by adding water to the mixture. The paste food was changed daily. The gel food that was supplemented with L-DOPA was created by dissolving 60 mg of L-DOPA and 15 mg of benserazide in 1 mL of ascorbic acid, prepared as described above, and adding it to the MediGel. We maintained the mice on this gel food for up to 3 days. We used both the paste food and gel food, depending on the specific experimental conditions.

### 4.3. Open-Field Test

Because the amount of food with L-DOPA could vary between mice, they received a subcutaneous injection of 50 mg/kg L-DOPA 3 days before the study, and they were given DietGel without L-DOPA for the remaining 3 days to standardize conditions during the test. This treatment regimen caused brain dopamine levels to fall below the limit of detection [15]. The open-field test (OFT) was performed by recording locomotion for 6 h using a Supermex apparatus (Muromachi Kikai, Tokyo, Japan). Each mouse was placed in an illuminated translucent chamber (350 mm × 400 mm × 250 mm). A sensor monitor was attached on top of the apparatus, and movements were automatically recorded and summed every 10 min. After a 3 h habituation period, the drugs were administered subcutaneously, and locomotor activity was monitored continuously for another 3 h. When two drugs were administered, WAY 100635 or scopolamine was administered 30 min before quetiapine administration. After the OFT, the mice were again treated with 50 mg/kg L-DOPA. Between the separate OFT sessions, the mice were allowed to rest for at least 2 weeks and maintained on L-DOPA-containing food.

### 4.4. Tissue Preparation

We conducted the OFT 72 h after the last 50 mg/kg L-DOPA injection. The drugs were administered 3 h after the start of the OFT, and brains were removed 1 h later (i.e., brains were removed 4 h after the start of the OFT). The mice were divided into three groups: no injection group (*n* = 5 WT, *n* = 7 DD), quetiapine group (*n* = 5 WT, *n* = 6 DD), and 8-OH-DPAT group (*n* = 6 WT, *n* = 6 DD). These mice were deeply anesthetized with pentobarbital, first refluxed with phosphate-buffered saline (PBS), and then fixed transcardially with 4% paraformaldehyde (PFA) using a perfusion pressure pump. The brain was removed and immersed in 4% PFA overnight. The solution was replaced with PBS and stored at 4 °C. Paraffin-embedded tissue sections (5 μm-thick) were cut with a sliding microtome.

### 4.5. Immunohistochemistry

The paraffin-embedded sections were deparaffinized, rehydrated, and immersed in distilled water. The sections were then autoclaved in 0.01 M citrate buffer (pH 6.0) for antigen activation. The sections were immersed in 0.3% hydrogen peroxide to remove endogenous peroxidase and treated with 5% normal goat serum for blocking. The sections were then stained with polyclonal rabbit anti-c-*fos* antibody (Synaptic Systems, Goettingen, Germany; 1:5000 dilution) for 3 days at 4 °C. Three days later, we stained the brain slices with biotin-linked goat anti-rabbit polyclonal antibody (Vector Laboratories, Burlingame, VT, USA; 1:200 dilution). We then stained the brain slices with avidin–biotin complex (Vector Laboratories; 1:100 dilution). We then applied 0.06% 3,3′-diaminobenzidine solution with 0.00012% hydrogen peroxide. Finally, the slides were dewatered, permeabilized, sealed, and observed by microscopy.

### 4.6. Cell Counting

Immunostained slides were photographed using an inverted fluorescence phase-contrast microscope (BZX800, Keyence, Osaka, Japan). Fos-positive neurons in the MRN and RLi were counted using the hybrid cell counting function in BZX800 analyzer software. The number of cells was counted in the range of 100 μm × 400 μm for the MRN and 200 μm × 200 μm for the RLi.

### 4.7. Statistical Analysis

We calculated the sample size based on an alpha error of 0.05, power of 0.9, and effect size of 0.8 using G*Power 3.1.9.7 [32]. Based on the results of the power analysis, we used more than 10 mice for each locomotor test. The statistical analysis was performed using two-way analysis of variance. Individual post hoc comparisons were performed using the Scheffé test. Sample normality and homogeneity were assessed using the Kolmogorov–Smirnov test and Leven’s test, respectively. Values of *p* < 0.05 were considered statistically significant. The data were analyzed using Bell-Curve for Microsoft Excel software (Social Survey Research Information 3.20, Tokyo, Japan).

## 5. Conclusions

In the present study, we found that quetiapine suppressed abnormal hyperlocomotion in DD mice. Quetiapine is often used clinically to treat PSDD, but its mechanism of action has not been clarified. The present findings suggest that 5-HT_1A_ receptors may mediate the therapeutic effects of quetiapine. An increase in activity in the MRN may be a pathological mechanism of PSDD.

## Figures and Tables

**Figure 1 ijms-23-07436-f001:**
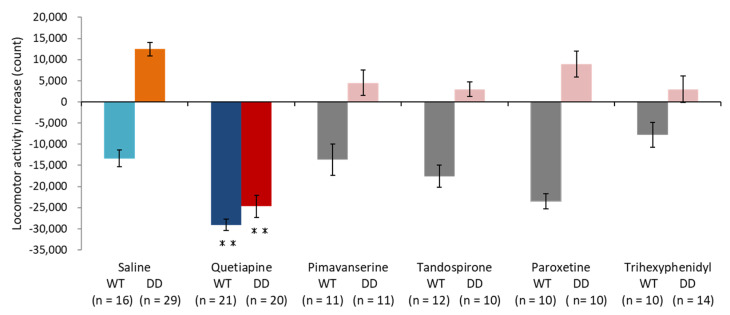
Behavioral changes related to PSDD induced by drug administration. Total locomotor activity increased (3 h after drug administration—3 h before drug administration). ** *p* < 0.01, compared with saline administration (Scheffé test). The graph shows the number of animals per group. The data are expressed as mean ± SEM. WT, wildtype mice; DD, dopamine-deficient mice.

**Figure 2 ijms-23-07436-f002:**
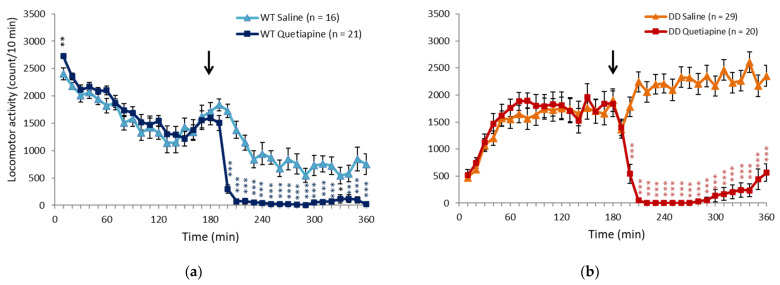
Changes in locomotor activity induced by 20 mg/kg quetiapine. (**a**,**b**) Changes in locomotor activity in (**a**) WT and (**b**) DD mice following quetiapine and saline administration. The arrows indicate the drug injection time. ** *p* < 0.01, *** *p* < 0.001, compared with saline (Student’s *t*-test). The graphs show the number of animals per group. The data are expressed as mean ± SEM. WT, wildtype mice; DD, dopamine-deficient mice.

**Figure 3 ijms-23-07436-f003:**
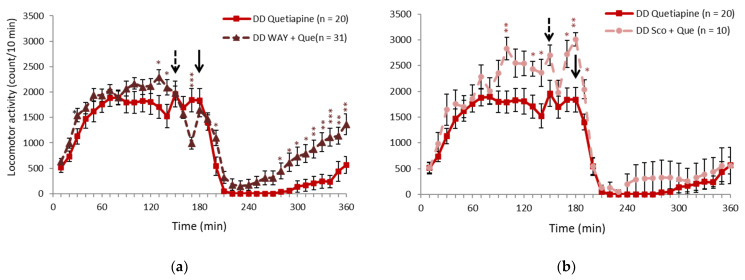
Effects of quetiapine on locomotor activity when 5-HT_1A_ and cholinergic receptor activity is suppressed. (**a**,**b**) Changes in locomotor activity induced by (**a**) 1 mg/kg WAY100635 and (**b**) scopolamine in DD mice 30 min before quetiapine administration. The solid arrows indicate the quetiapine injection time. The dashed arrows indicate the WAY100635 and scopolamine injection time. * *p* < 0.05, ** *p* < 0.01, *** *p* < 0.001, compared with quetiapine (Student’s *t*-test). The graphs show the number of animals per group. The data are expressed as mean ± SEM. 5-HT, 5-hydroxytryptamine (serotonin); DD, dopamine-deficient mice.

**Figure 4 ijms-23-07436-f004:**
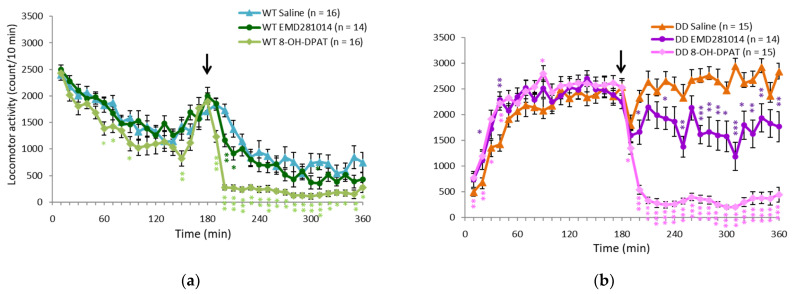
Different effects of 5-HT and 5-HT_2A_ receptor activation on hyperactivity in DD mice. (**a**,**b**) Changes in locomotor activity in (**a**) WT and (**b**) DD mice after EMD281014 (5-HT_2A_ receptor antagonist) and 8-OH-DPAT (5-HT_1A_ receptor agonist) administration. The arrows indicate the drug administration time. * *p* < 0.05, ** *p* < 0.01, *** *p* < 0.001, compared with saline (Student’s *t*-test). The graphs show the number of animals per group. The data are expressed as mean ± SEM. WT, wildtype mice; DD, dopamine-deficient mice; 8-OH-DPAT, 8-hydroxy-2-dipropylaminotetralin hydrobromide.

**Figure 5 ijms-23-07436-f005:**
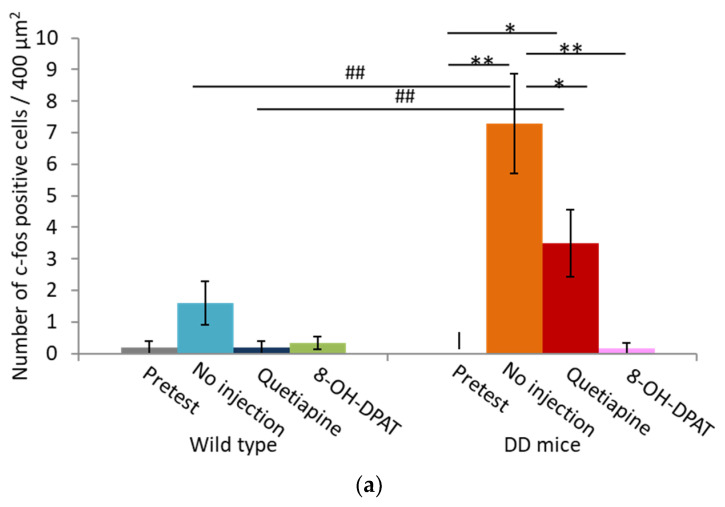
Neuronal activity after quetiapine and 8-OH-DPAT administration in the median raphe nucleus in DD mice. (**a**) number of Fos-positive cells per 0.04 mm^2^ in the median raphe nucleus before and 1 h after quetiapine and 8-OH-DPAT administration in WT and DD mice. * *p* < 0.05, ** *p* < 0.01, compared with no injection (Scheffé test); ## *p* < 0.01, compared with WT (Scheffé test). The data are expressed as mean ± SEM. (**b**) Representative immunohistochemical images before and 1 h after quetiapine and 8-OH-DPAT administration in WT and DD mice. Scale bar = 100 µm. Fos-positive cells are indicated by arrows. The blue line depicts the shape of the MRN. 8-OH-DPAT, 8-hydroxy-2-dipropylaminotetralin hydrobromide; WT, wildtype mice; DD, dopamine-deficient mice.

**Figure 6 ijms-23-07436-f006:**
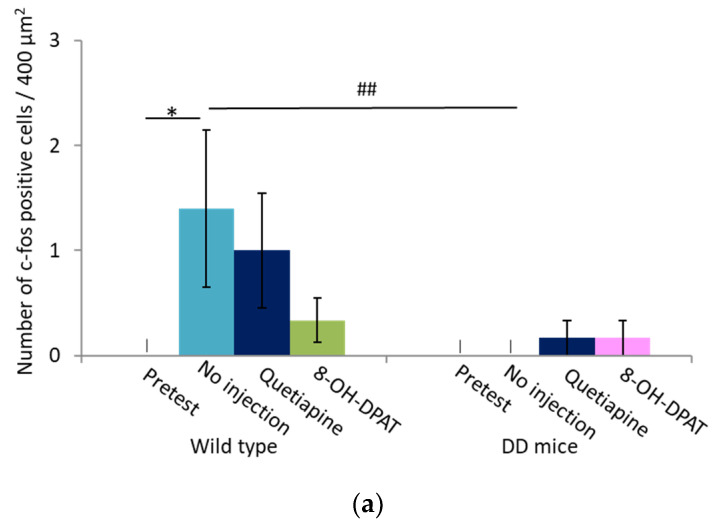
Neuronal activity after quetiapine and 8-OH-DPAT administration in the rostral linear nucleus in DD mice. (**a**) Number of Fos-positive cells per 0.04 mm^2^ of the rostral linear nucleus before and 1 h after quetiapine and 8-OH-DPAT administration in WT and DD mice. * *p* < 0.05, compared with no injection (Scheffé test); ## *p* < 0.01, compared with WT (Scheffé test). The data are expressed as mean ± SEM. (**b**) Representative immunohistochemical images before and 1 h after quetiapine and 8-OH-DPAT administration in WT and DD mice. Scale bar = 100 µm. Fos-positive cells are indicated by arrows. The blue line depicts the shape of the RLi. 8-OH-DPAT, 8-hydroxy-2-dipropylaminotetralin hydrobromide; WT, wildtype mice; DD, dopamine-deficient mice.

## Data Availability

All data are included in the dataset as a Appendix A.

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
