# Peer review of "Therapeutic Effects of Quetiapine and 5-HT1A Receptor Agonism on Hyperactivity in Dopamine-Deficient Mice"

_ijms, 2022, doi:10.3390/ijms23137436_

Round 1

Reviewer 1 Report

In the manuscript entitled “Therapeutic effects of quetiapine and 5-HT1A receptor agonism on hyperactivity in dopamine-deficient mice”, Yukiko Ochiai et al demonstrated the therapeutic effect of 5HT1A agonists and quetiapine on the hyperactive phenotype of dopamine-deficient mice. This is an interesting manuscript and, overall the strategy used by the authors is clear and the manuscript is well written.

I have a few concerns and suggestions that I believe would enrich the article:

- The hyperactivity described by the authors is somehow confusing to me, as the phenotypes described (jumps, running around, etc) are more characteristic of stereotypic behavior. Definition of Hyperactivity in mice should be better supported with literature in the introduction section, as the study presented here is based on this hyperactivity phenotype of DD mice.

- I worry about the L-DOPA supplementation in the format of paste/gel food. How can the authors assure that all the animals eat equal amounts of food? This would mean a lot of variability among the dopamine-deficient mice. This should be further discussed.

- The statistical section must be improved. Please report how sample normality and homogeneity were assessed. Furthermore, sample size calculations should be explained.

In my perspective, the statistical analysis performed in figure 1 is not appropriate. Multiple comparisons should be applied.

- The higher locomotor decrease by quetiapine treatment in WT mice should be discussed (comparing to baseline).

 - Figures 5 b) and 6 b) quality must be improved or higher magnifications can be used; it’s really hard to see what the authors are showing in the quantification graph.

 - The discussion section can be improved, as few studies on therapies using DD animals were discussed.

Author Response

Reviewer 1

In the manuscript entitled “Therapeutic effects of quetiapine and 5-HT1A receptor agonism on hyperactivity in dopamine-deficient mice”, Yukiko Ochiai et al demonstrated the therapeutic effect of 5HT1A agonists and quetiapine on the hyperactive phenotype of dopamine-deficient mice. This is an interesting manuscript and, overall the strategy used by the authors is clear and the manuscript is well written.

I have a few concerns and suggestions that I believe would enrich the article:

- The hyperactivity described by the authors is somehow confusing to me, as the phenotypes described (jumps, running around, etc) are more characteristic of stereotypic behavior. Definition of Hyperactivity in mice should be better supported with literature in the introduction section, as the study presented here is based on this hyperactivity phenotype of DD mice.

Authors’ Response:

We thank the reviewer for this insightful advice. We added “which is detected by an increase in locomotor activity.” We also added the word “sometimes” after “jump” because it is not a stereotypic behavior. We also added “without becoming acclimated to their environment.”

(page 2, lines 65-69, Introduction)

- I worry about the L-DOPA supplementation in the format of paste/gel food. How can the authors assure that all the animals eat equal amounts of food? This would mean a lot of variability among the dopamine-deficient mice. This should be further discussed.

Authors’ Response:

As the reviewer mentioned, we cannot guarantee that all animals were eating the same amount of food outside the experimental preparation period. We added the following text:

“Because the amount of food with L-DOPA could vary between mice, they received a subcutaneous injection of 50 mg/kg L-DOPA 3 days before the study, and they were given DietGel without L-DOPA for the remaining 3 days to standardize conditions during the test. This treatment regimen caused brain dopamine levels to fall below the limit of detection [15].”

(page 10, lines 321-326, Methods)

- The statistical section must be improved. Please report how sample normality and homogeneity were assessed. Furthermore, sample size calculations should be explained.

Authors’ Response:

We thank the reviewer for this instructive comment. Sample normality, homogeneity, and sample size calculations were added:

“We calculated the sample size based on an alpha error of 0.05, power of 0.9, and effect size of 0.8 using G*Power 3.1.9.7 [32]. Based on the results of the power analysis, we used more than 10 mice for each locomotor test. The statistical analysis was performed using two-way analysis of variance. Individual post hoc comparisons were performed using the Scheffe test. Sample normality and homogeneity were assessed using the Kolmogorov-Smirnov test and Leven’s test, respectively.”

(page 11, lines 366-371, Methods)

-  In my perspective, the statistical analysis performed in figure 1 is not appropriate. Multiple comparisons should be applied.

Authors’ Response:

As the reviewer indicates, there was an error in our description. We replaced “Student’s t-test” with “Scheffe test.”

(page 3, line 118, Figure 1 Legend)

- The higher locomotor decrease by quetiapine treatment in WT mice should be discussed (comparing to baseline).

Authors’ Response:

We added the following as a limitation:

“Additionally, both DD mice and wildtype mice exhibited a decrease in locomotor activity in the novel environment when they were treated with quetiapine and 8-OH-DPAT. Therefore, unknown is whether quetiapine and 8-OH-DPAT specifically suppressed hyperlocomotion in DD mice.”

(page 9, lines 267-270, Discussion)

- Figures 5 b) and 6 b) quality must be improved or higher magnifications can be used; it’s really hard to see what the authors are showing in the quantification graph.

Authors’ Response:

We agree with the reviewer that the figures were unclear. We changed the format of the figure file from TIF to PDF. We also added arrows to the figures so that readers can easily understand which cells are Fos-positive.

(pages 6 and 7, Figures 5b, 6b)

We also added the following explanation to the figure legend:

“Fos-positive cells are indicated by arrows. The blue line depicts the shape of the MRN.”

(page 6, lines 188-189, Figure 5 legend)

“Fos-positive cells are indicated by arrows. The blue line depicts the shape of the RLi.”

(page 7, lines200-201 , Figure 6 legend)

- The discussion section can be improved, as few studies on therapies using DD animals were discussed.

Authors’ Response:

We thank the reviewer for this instructive comment. We added the following paragraph:

“Psychiatric symptoms have been studied in other DD animals. DD animals that exhibit psychiatric symptoms include 1-methyl-4-phenyl-1,2,3,6-tetrahydropyridine (MPTP)-lesioned marmosets [27], human a-synuclein transgenic rats [18], and 6-hydroxydopamine-lesioned rats [28]. However, psychotic behaviors appeared when these animals received drugs that stimulate dopamine receptors. Therefore, these models may not be suitable for studying the pathogenesis of psychosis in a dopamine-depleted state. With regard to 5-HT and psychosis, L-DOPA-stimulated psychosis-like behavior in MPTP-lesioned monkeys was abolished after 5-HT injury by 3,4-methylenedioxy-N-methamphetamine (MDMA) [29]. Although the animals in this previous study received L-DOPA, the fact that serotonergic neuron injury can improve neuropsychiatric-like behaviours in DD animals appears to support the present results, in which a decrease in serotonergic signaling could ameliorate psychotic-like behavior in DD animals.”

(page 8, lines 242-253, Discussion)

Reviewer 2 Report

Review of a manuscript: ”Therapeutic effects of quetiapine and 5-HT1A receptor agonism on hyperactivity in dopamine-deficient mice” by Yukiko Ochiai and coauthors submitted to IJMS.

Association of dopamine deficiency with psychiatric symptoms is an important and not completely understood biomedical issue. To study the mechanism of this association the authors used dopamine-deficient mice that exhibited hyperactivity in a novel environment. The area of this research is important and the results presented in the manuscript will be interesting for the readers of IJMS.

The following corrections and additions should be made:

Abstract

Line 13. “Some diseases that are associated with dopamine deficiency are accompanied by psychiatric symptoms, such as Parkinson’s disease.” The sentence begins in plural (diseases) but the authors give as example only one (Parkinson’s disease ). The authors should either give oother examples or rewrite the sentence replaceing “such as” on “including”.

Lines 22-23: ”Quetiapine suppressed hyperactivity in DD mice. The 5-HT1A receptor antagonist WAY100635 inhibited this effect.” The sentences should be combined as “Quetiapine suppressed hyperactivity in DD mice while the 5-HT1A receptor antagonist WAY100635 inhibited this effect.”

 Introduction

Lines 39-41: ”Dopamine levels also decrease in Parkinson’s disease (PD) through the degeneration of dopaminergic neurons in the substantia nigra, the cause of which has not yet been clarified.” The authors should add here a citation of a recent review on Parkinson’s disease “”Biomarkers in Parkinson’s Disease”. Chapter in a book Peplow P.V., Martinez B., Gennarelli T.A. (eds) Neurodegenerative Diseases Biomarkers. 2022. Neuromethods, vol 173. pp 155-180. Humana, New York, NY. https://link.springer.com/protocol/10.1007/978-1-0716-1712-0_7

 Lines 51,52 and 57: the abbreviations “psychiatric symptoms associated with dopamine deficiency (PSDD)” and  “dopamine-deficient (DD) mice” were already used above, so there is no need to repeat them again.

Line 71:” PSDD and hyperactivity in DD mice occurs…” should be corrected as “PSDD and hyperactivity in DD mice occur…”

 Results

Line 87: “We first examined the effects of drugs that influence hyperactivity in DD mice.“The authors should be more specific indicating what kind of effects they mean. “Effects on what?”

Discussion

In discussion the authors should add brief arguments how their results may be used for translational application.  

 Lines 198-199: “Quetiapine …effectively ameliorated hyperactivity” The word “ameliorated” should be replaced on another term which is more specific.

 Lines 240-241. The authors should discuss in more details possible drawbacks of using c-Fos. What mistakes can be caused by using an indirect marker of neuronal activity results?  

.

Author Response

Reviewer 2

Association of dopamine deficiency with psychiatric symptoms is an important and not completely understood biomedical issue. To study the mechanism of this association the authors used dopamine-deficient mice that exhibited hyperactivity in a novel environment. The area of this research is important and the results presented in the manuscript will be interesting for the readers of IJMS.

The following corrections and additions should be made:

Abstract

Line 13. “Some diseases that are associated with dopamine deficiency are accompanied by psychiatric symptoms, such as Parkinson’s disease.” The sentence begins in plural (diseases), but the authors give as example only one (Parkinson’s disease). The authors should either give other examples or rewrite the sentence replaceing “such as” on “including”.

Authors’ Response:

We thank the reviewer for the helpful comments. As the reviewer suggested, we replaced “such as” with “including.”

(page 1, lines 14, Abstract)

Lines 22-23: ”Quetiapine suppressed hyperactivity in DD mice. The 5-HT1A receptor antagonist WAY100635 inhibited this effect.” The sentences should be combined as “Quetiapine suppressed hyperactivity in DD mice while the 5-HT1A receptor antagonist WAY100635 inhibited this effect.”

Authors’ Response:

We fixed the text in the Abstract:

“Quetiapine suppressed hyperactivity in DD mice while the 5-HT1A receptor antagonist WAY100635 inhibited this effect.”

(page 1, lines 22-23, Abstract)

Introduction

Lines 39-41: ”Dopamine levels also decrease in Parkinson’s disease (PD) through the degeneration of dopaminergic neurons in the substantia nigra, the cause of which has not yet been clarified.” The authors should add here a citation of a recent review on Parkinson’s disease “”Biomarkers in Parkinson’s Disease”. Chapter in a book Peplow P.V., Martinez B., Gennarelli T.A. (eds) Neurodegenerative Diseases Biomarkers. 2022. Neuromethods, vol 173. pp 155-180. Humana, New York, NY. https://link.springer.com/protocol/10.1007/978-1-0716-1712-0_7

Authors’ Response:

We thank the reviewer for sharing this article with us. We added it to the references (no. 5).

Lines 51,52 and 57: the abbreviations “psychiatric symptoms associated with dopamine deficiency (PSDD)” and “dopamine-deficient (DD) mice” were already used above, so there is no need to repeat them again.

Authors’ Response:

This was corrected.

(page 2, lines 51, 52, and 57, Introduction)

Line 71:” PSDD and hyperactivity in DD mice occurs…” should be corrected as “PSDD and hyperactivity in DD mice occur…”

Authors’ Response:

We changed this to “occur.”

(page 2, lines 73, Introduction)

Line 87: “We first examined the effects of drugs that influence hyperactivity in DD mice.“The authors should be more specific indicating what kind of effects they mean. “Effects on what?”

Authors’ Response:

We appreciate the reviewer’s comment. We specifically stated the effect as the following:

We first examined the effects of drugs on hyperlocomotion in DD mice.

(page 2, lines 89-90, Results)

Discussion

In discussion the authors should add brief arguments how their results may be used for translational application.

Authors’ Response:

We thank the reviewer for this comment. We added a sentence to the Discussion:

“In addition to providing clues to the mechanism of action of quetiapine on PSDD, 5-HT1A receptors may be a target for effectively treating PSDD.”

(page 8, lines 216-218, Discussion)

Lines 198-199: “Quetiapine …effectively ameliorated hyperactivity” The word “ameliorated” should be replaced on another term which is more specific.

Authors’ Response:

We replaced “ameliorated” with “suppressed.”

(page 7, lines 206, Discussion)

Lines 240-241. The authors should discuss in more details possible drawbacks of using c-Fos. What mistakes can be caused by using an indirect marker of neuronal activity results? 

Authors’ Response:

As the reviewer mentioned, we should have further discussed the drawbacks of using c-Fos. We added the following sentence:

“The appearance of c-Fos-positive cells that is attributable various nonspecific causes should also be considered.”

(page 9, lines 264-266, Discussion)
